# Identification of the Multiresistance Gene *poxtA* in Oxazolidinone-Susceptible *Staphylococcus haemolyticus* and *Staphylococcus saprophyticus* of Pig and Feed Origins

**DOI:** 10.3390/pathogens10050601

**Published:** 2021-05-14

**Authors:** Lin Chen, Jian-Xin Hu, Chang Liu, Jiao Liu, Zhen-Bao Ma, Zi-Yun Tang, Ya-Fei Li, Zhen-Ling Zeng

**Affiliations:** 1College of Veterinary Medicine, Guangdong Provincial Key Laboratory of Veterinary Pharmaceutics Development and Safety Evaluation, National Risk Assessment Laboratory for Antimicrobial Resistance of Animal Original Bacteria, South China Agricultural University, Guangzhou 510642, China; chenl@stu.scau.edu.cn (L.C.); 20192027012@stu.scau.edu.cn (J.-X.H.); 20182027011@stu.scau.edu.cn (C.L.); 20182027012@stu.scau.edu.cn (J.L.); mazhenbao@stu.scau.edu.cn (Z.-B.M.); 20182027026@stu.scau.edu.cn (Z.-Y.T.); 2Guangdong Laboratory for Lingnan Modern Agriculture, South China Agricultural University, Guangzhou 510642, China; 3Public Monitoring Center of Agro-Product of Guangdong Academy of Sciences, Guangzhou 510640, China

**Keywords:** *poxtA*, CoNS, transformation, plasmids, antimicrobial, heavy metal, IS*1216*

## Abstract

Previous studies on the prevalence and transmission mechanism of oxazolidinone resistance gene *poxtA* in CoNS are lacking, which this study addresses. By screening 763 CoNS isolates from different sources of several livestock farms in Guangdong, China, 2018–2020, we identified that the *poxtA* was present in seven CoNS isolates of pig and feed origins. Species identification and multilocus sequence typing (MLST) confirmed that seven *poxtA*-positive CoNS isolates were composed of five ST64-*Staphylococcus haemolyticus* and two *Staphylococcus saprophyticus* isolates. All *poxtA*-positive *Staphylococcus haemolyticus* isolates shared similar pulsed-field gel electrophoresis (PFGE) patterns. Transformation assays demonstrated all *poxtA*-positive isolates were able to transfer *poxtA* gene to *Staphylococcus aureus* RN4220. S1-PFGE and whole-genome sequencing (WGS) revealed the presence of *poxtA*-carrying plasmids in size around 54.7 kb. The plasmid pY80 was 55,758 bp in size and harbored the heavy metal resistance gene *czcD* and antimicrobial resistance genes, *poxtA*, *aadD*, *fexB* and *tet*(L). The regions (IS*1216E*-*poxtA*-IS*1216E*) in plasmid pY80 were identified in *Staphylococcus* spp. and *Enterococcus* spp. with different genetic and source backgrounds. In conclusion, this was the first report about the *poxtA* gene in *Staphylococcus haemolyticus* and *Staphylococcus saprophyticus*, and IS*1216* may play an important role in the dissemination of *poxtA* among different Gram-positive bacteria.

## 1. Introduction 

Coagulase negative staphylococci (CoNS) are one of the most common opportunistic pathogens found on human skin and mucous membranes as a component of normal flora [1,2]. Besides their role in keeping homeostasis, CoNS have been involved in a series of infectious processes, ranging from nosocomial infections to livestock bacterial sepsis and mastitis [3,4]. In addition to their virulence, the emergence of antibiotic resistance in CoNS and then horizontal dissemination among staphylococci should be alarming. The increasing drug resistance of CoNS significantly limited the treatment options [5,6]. Among CoNS, *Staphylococcus haemolyticus* is the second most frequently isolated from human blood culture and *Staphylococcus saprophyticus* is one of the most common pathogens responsible for community urinary tract infections [5,7].

Oxazolidinones such as linezolid and tedizolid are antibacterially active against Gram-positive pathogens including methicillin-resistant *Staphylococcus aureus* (MRSA), methicillin-resistant CoNS and vancomycin-resistant enterococci (VRE) [8]. However, the discovery of transferable oxazolidinone resistance genes such as *cfr*, *cfr*(B), *cfr*(C), *optrA* and *poxtA* as well as the mutations in 23S rRNA and ribosomal proteins L3 and L4 challenged the clinical use of oxazolidinones [9]. Worriedly, linezolid-resistant staphylococci have been detected worldwide [10]. The plasmid-mediated oxazolidinone resistance genes including *cfr* and *optrA* spread among a number of bacterial species of different origins around the world shortly after they were reported [11]. The fact that *cfr* and *optrA* genes can be selected by phenicols and other ribosomal-targeted drugs that are widely used in livestock and veterinary hospitals is closely associated with global spread of the resistance genes [12,13,14]. The recently described plasmid-mediated oxazolidinone resistance gene *poxtA* could decrease susceptibility to phenicols and tetracyclines, so the *poxtA* gene posed a threat to disseminate in bacteria from animal setting [15,16]. The *poxtA* gene has been identified in MRSA and *Enterococcus* strains of human and animal origins [15,16,17,18]. Livestock is widely recognized as a reservoir of antimicrobial resistance genes [19].

In this study, we described for the first time the identification and characterization of *poxtA* gene in *S. haemolyticus* and *S. saprophyticus* isolates from pig and chicken farms in Guangdong province, China.

## 2. Results

### 2.1. Identification of poxtA Gene in CoNS Isolates

The *poxtA* gene was detected in seven CoNS isolates in 2018 including five *S. haemolyticus* isolates (GDY8P33P, GDY8P50P, GDY8P58P, GDY8P60P and GDY8P80P all of pig origin) and two *S. saprophyticus* isolates (GDY8P136P of pig origin, GDH8C97P of feed origin) (Table 1).

### 2.2. Distribution of ARGs in poxtA-Positive CoNS Isolates and the Electrotransformants

In total, 16 additional ARGs were detected among the *poxtA*-positive CoNS isolates (Figure 1). ARGs were widespread in the *poxtA*-positive isolates of both feed and pig origins. Except for the widely distributed resistance genes *aadD*, *ant(6)-Ia*, *blaZ*, *mecA*, *lsa*(E), *lnu*(B), *erm*(C), *fexB*, *tet*(L) and *dfrG* in the *poxtA*-positive CoNS isolates, the distribution of individual ARG varied among the *poxtA*-positive CoNS isolates. For example, the *cfr*, *tet*(M) and *aac(6*′*)-aph(2″)* genes were also identified in the *poxtA*-positive *S. haemolyticus* isolates (Figure 1). All *poxtA*-positive CoNS isolates were able to transfer the *fexB*, *poxtA* and *tet*(L) genes to *S. aureus* strain RN4220 (Table 2). In addition, the five *poxtA*-positive *S. haemolyticus* isolates were able to transfer the *aadD* gene to *S. aureus* strain RN4220 (Table 2).

### 2.3. Antimicrobial Susceptibility

Antimicrobial susceptibility testing showed that resistance rates of seven *poxtA*-positive CoNS isolates to penicillin, cefoxitin, doxycycline, tetracycline, florfenicol, erythromycin and ciprofloxacin reached 100% (Table 2). Six (85.7%) *poxtA*-positive CoNS isolates demonstrated resistance to enrofloxacin. All *poxtA*-positive CoNS isolates remained susceptible to linezolid, tedizolid, tigecycline, amikacin, gentamicin, rifampicin and vancomycin. All *poxtA*-positive CoNS isolates were able to transfer the florfenicol resistance to *S. aureus* strain RN4220. Two electrotransformants are erythromycin resistant (Table 2). In addition, the electrotransformants carrying *aadD*, *poxtA* and *tet*(L) genes exhibited lower MICs of neomycin, kanamycin, linezolid, tedizolid, doxycycline and tetracycline compared with the donors (Table 2).

### 2.4. Phylogenetic Relatedness of poxtA-Positive CoNS Isolates

All *poxtA*-positive *S. haemolyticus* isolates derived from swine nasal swabs in pig farm D represented ST64 by MLST analysis and were closely related by phylogenetic analysis (Table 1 and Figure 1). In addition, 121 SNPs were identified in all *poxtA*-positive *S. haemolyticus* isolates (Figure 1). The *S. saprophyticus* isolates GDH8C97P recovered from feed in chicken farm A and GDY8P136P recovered from swine nasal swab in pig farm D shared 1957 SNPs difference (Figure 1).

### 2.5. Plasmids Analysis

S1-PFGE and WGS analysis confirmed that the *poxtA* gene in the seven CoNS isolates and corresponding electrotransformants was located on plasmids ranging in size around 54.7 kb (Figure 2 and Appendix A). Plasmid pY80 carrying the *poxtA* gene was 55,758 bp in size and exhibited <38% coverage with other plasmids in NCBI database, with an average GC content of 34.0%. In total, 51 ORFs coding for proteins of >50 amino acids were identified (Figure 2). Except for the 14 ORFs encoding hypothetical proteins with no defined function, the products of the remaining 37 ORFs exhibited identities ranging from 76.3% to 100% to proteins with known functions, including antimicrobial resistance, heavy metal resistance, conjugative transfer or transposition, plasmid replication and other function (Figure 2).

### 2.6. Genetic Environment of poxtA Gene

The *poxtA*-carrying segments (IS*1216*-*poxtA*-IS*1216*-*fexB*-IS*431mec*-*tet*(L)-*aadD*-IS*431mec*) of 17287 bp in plasmid pY80 of pig origin) were selected to conduct comparative analysis with other *poxtA*-carrying segments. The IS*1216*-*poxtA*-IS*1216* segment of 4130 bp showed >98% identity to corresponding sequences in two *Enterococcus hirae* plasmids (pHDC14-2.27K and pfas4-1 both of pig origins), two *Enterococcus faecalis* plasmids (pM18/0011 of human origin and pC10 of pig origin), 10 *Enterococcus faecium* plasmids (pSDGJQ5 of chicken origin, pM160954 of human origin, pE1077-23 of pig origin, pSCBC1 of pig origin, pSDGJP3 of pig origin, pYN2-1 of pig origin, pHN11 of pig origin, pGZ8 of pig origin, pSC3-1 of chicken origin and pC25-1 of pig origin) and the genome of *S. aureus* AOUC-0915 of human origin (Figure 3). In addition, the IS*1216*-*poxtA* segment of 2363 bp showed >98% identity to corresponding sequences in the genomes of *Enterococcus faecium* P36 of pig origin and *Pediococcus acidilactici* BCC1 of chicken origin (Figure 3). The 11951 bp region (*fexB*-IS*431mec*-*tet*(L)-*aadD*-IS*431mec*) downstream of IS*1216*-*poxtA*-IS*1216* in plasmid pY80 was identified. Within this region, the 4200 bp segment (*tet*(L)-*aadD*-IS*431mec*) showed >98% identity to the *S. aureus* plasmid (pERGB of human origin) (Figure 3). The *poxtA*-carrying fragments that often harbor additional resistance genes such as *fexB*, *tet*(L) and *tet*(M) were identified in different bacterial species (Figure 3).

## 3. Discussion

CoNS are recognized as significant opportunistic pathogens that cause infections in humans and animals [4,5], and CoNS carrying important antimicrobial resistance genes such as oxazolidinone resistance genes could pose a huge burden on the healthcare system and breeding industry [10]. The transferable oxazolidinone resistance gene *poxtA* in different enterococci was the most recently reported [17]. Attention should be paid to the fact that the *poxtA* gene was originally detected in a linezolid-resistant MRSA strain [16]. Therefore, there was a risk that the *poxtA* could spread to other bacterial strains. The observation that the *poxtA* gene was mainly detected in *S. haemolyticus* and *S. saprophyticus* isolates might suggest an *S. haemolyticus* and *S. saprophyticus* reservoir. In this study, the results indicated that *poxtA*-positive ST64-*S. haemolyticus* isolates from swine nasal swabs in a pig farm shared low SNPs difference and were closely related, and that *poxtA*-positive *S. saprophyticus* isolates from a pig farm and a chicken farm shared a high SNPs difference. Therefore, ST64-*S. haemolyticus* isolates carrying *poxtA* can spread among pigs in the pig farm, and the potential spread of *S. saprophyticus* isolates carrying *poxtA* between the pig farm and chicken farm should arouse people’s attention.

It was reported that IS*1216* played a major role in the processes of aiding the dissemination and persistence of *poxtA* among enterococci [20]. The presence of these homologous gene regions (IS*1216*-*poxtA*-IS*1216* or IS*1216*-*poxtA*) in *Staphylococcus* spp., *Enterococcus* spp. and *Pediococcus* spp. confirmed that IS*1216* was closely related to the spread of *poxtA* among these Gram-positive bacteria with different genetic and source backgrounds. This was a further reminder that the *poxtA* gene might have spread widely in these bacteria. It has been found from reported studies that most of these homologous gene regions are located on transferable plasmids which often harbor additional resistance genes such as the tetracycline resistance genes *tet*(M) and *tet*(L), and the phenicol exporter gene *fexB* [15,16,17,18]. In this study, plasmid pY80 carried heavy metal resistance gene *czcD* and aminoglycoside-modifying enzyme gene *aadD* in addition to *tet*(L), *fexB* and *poxtA* genes. The co-occurrence of *poxtA* with other antimicrobial and heavy metal resistance genes on the transferable plasmids may lead to the co-selection of *poxtA*, contributing to its persistence and accelerating its dissemination [15]. The *poxtA* gene was identified in the new plasmid. Once the *poxtA* gene is inserted into a plasmid with strong transmission ability, it will bring great difficulties to control the further transmission of the *poxtA* gene. In China, florfenicol has been widely used in food-producing animals [21]. It was reported that the emergence of oxazolidinone resistance genes such as *poxtA* is closely related to the use of florfenicol in breeding farms [22]. Antimicrobial susceptibility testing showed that all donors and electrotransformants were resistant to florfenicol, indicating *poxtA*-positive plasmids could be directly selected by florfenicol. The phenomenon that the strains and electrotransformants investigated in this study were phenotypically oxazolidinone-susceptible despite the fact that they carry up to two oxazolidinone resistance genes (*cfr* and *poxtA*) is very interesting. This may be due to the possibility that *cfr* was not transcribed [23] and *poxtA* played a relatively low role on oxazolidinone susceptibility [16]. The fact that electrotransformants with *aadD* and *tet*(L) genes did not show resistance to kanamycin, neomycin, doxycycline and tetracycline might be related to the silencing of these genes. It is easy for people to ignore the resistance genes without corresponding drug-resistant phenotypes, resulting in the widespread spread of them [24]. That antimicrobial agents used in livestock could exert selective pressures on bacteria [25] and all the *poxtA*-positive CoNS isolates exhibited multidrug resistance and carried additional resistance genes should account for the spread of *poxtA* gene in the CoNS isolates [15].

## 4. Materials and Methods

### 4.1. Bacterial Isolations and Detection of poxtA Gene

A total of 778 CoNS isolates were collected from 1 chicken farm, 15 pig farms and 18 duck farms in Guangdong, China, between 2018–2020. Isolates were recovered from 34.6% (9/26) of human nasal swabs, 58.9% (353/599) of swine nasal swabs, 39.8% (33/83) of feed samples, 47.5% (94/198) of pond water samples, 34.3% (35/102) of soil samples, 65.4% (51/78) of airborne dust samples and animal 52.3% (203/388) of viscera samples.

All isolates were screened for the presence of *poxtA* by PCR using previously described primers [15]. Species identification was performed using MALDI-TOF MS (Bruker Daltonik GmbH, Bremen, Germany) and further confirmed by 16S rDNA sequence analysis.

### 4.2. Molecular Epidemiology Analysis and Transformation Assays

Multilocus sequence typing (MLST) was conducted for identification of clonal correlation of the *poxtA*-positive *S. haemolyticus* (http://www.shaemolyticus.mlst.net Accessed on: 29 January 2021) [26]. Plasmid DNA from all *poxtA*-positive CoNS isolates was extracted using a Qiagen Prep Plasmid Midi Kit (Qiagen, Hilden, Germany) and transferred into a recipient *S. aureus* strain RN4220 by electroporation using Gene Pulser apparatus (Bio-Rad, Hercules, CA, United States) [27]. Electrotransformants were selected on brain heart infusion (BHI) agar containing 10 µg/mL of florfenicol. Electrotransformants were further confirmed for the presence of *poxtA* gene by PCR analysis. The successful electrotransformants were further screened for the presence of *aadD*, *fexB*, *tet*(L) and *tet*(M) genes by PCR.

### 4.3. Antimicrobial Susceptibility Testing

All *poxtA*-positive CoNS isolates and corresponding electrotransformants were investigated for their MICs of florfenicol, linezolid, tedizolid, amoxicillin, penicillin, cefoxitin, doxycycline, tetracycline, tigecycline, gentamicin, amikacin, neomycin, kanamycin, erythromycin, ciprofloxacin, enrofloxacin, rifampicin, vancomycin and trimethoprim-sulfamethoxazole by broth microdilution following the recommendations given in CLSI documents VET01-S2 and M100-S30 [28,29]. *Staphylococcus aureus* ATCC 29213 was used as the quality control strain. 

### 4.4. S1-PFGE and WGS Analysis

Genomic DNA of all *poxtA*-positive CoNS isolates and corresponding electrotransformants were digested with S1 endonuclease and separated by PFGE as previously described [30]. Whole-cell DNA of all *poxtA*-positive CoNS isolates were prepared using the HiPure Bacterial DNA Kit (Magen, Guangzhou, China), following the manufacturer’s instructions, and then preceded by library construction on Novaseq 6,000 sequencing platform, which produced 150 bp paired-end reads (Novogene Company, Beijing, China). Novaseq sequences were assembled using CLC Genomics Workbench 10 (CLC Bio, Aarhus, Denmark). The GDY8P80P isolate carrying *poxtA* and *cfr* genes was further used for whole-genome sequencing on PacBio RS II sequencing platform (Biochip Company, Tianjin, China). Pacbio sequences were assembled using hierarchical genome-assembly process [31]. The assembled Pacbio sequences were corrected through Burrows–Wheeler Aligner’s Smith–Waterman Alignment (BWA-SW) software to ensure their integrity according to Novaseq sequences [32]. The plasmids carrying *poxtA* were annotated using the Rapid Annotation of microbial genomes using Subsystems Technology annotation server (http://rast.nmpdr.org/ Accessed on: 29 January 2021) [33]. Acquired resistance genes (ARGs) were identified in the genomes using ResFinder 4.0 [34]. The genetic comparison of the *poxtA* gene from different species was generated using Easyfig 2.1 [35]. Based on the draft genome sequences, a phylogenic tree was constructed for all sequenced *poxtA*-positive CoNS isolates by CSI Phylogeny 1.4 (https:// cge.cbs.dtu.dk/services/CSIPhylogeny/ Accessed on: 29 January 2021), with the genome of GDY8P50P used as a reference. The tree was visualized using software Fig Tree 1.4.2. Single nucleotide polymorphism (SNP) divergence among various isolates carrying *poxtA* was calculated using snippy (https:// www.github.com/heilaaks/snippy/ Accessed on: 29 January 2021).

### 4.5. Nucleotide Sequence Accession Numbers

The complete sequences of strains GDH8C97P, GDY8P33P, GDY8P50P, GDY8P58P, GDY8P60, GDY8P80P and GDY8P136P have been deposited in GenBank under accession numbers JADICE000000000, JADQVZ000000000, JADICF000000000, JADICG000000000, JADICH000000000, JADICI000000000 and JADICJ000000000, respectively. The complete sequence of plasmid pY80 have been deposited in GenBank under accession numbers CP063444.

### 4.6. Ethical Considerations

The study was approved by the South China Agriculture University (SCAU) Animal Ethics Committee. The research was conducted in strict accordance with Section 20 of the Animal Diseases Act of 1984 (Act No 35 of 1984) and the Declaration of Helsinki, and was approved by the SCAU Institutional Animal Care and Use Committee.

## 5. Conclusions

In conclusion, this is the first study to report on the presence of the *poxtA* gene in livestock-derived *S. haemolyticus* and *S. saprophyticus*. The presence of IS*1216*-*poxtA*-IS*1216* in *Staphylococcus* spp., *Enterococcus* spp. and *Pediococcus* spp. with different genetic and source backgrounds indicated an important role of IS*1216* in the dissemination of *poxtA*. Moreover, the co-occurrence of *poxtA* with other antimicrobial and heavy metal resistance genes on the transferable plasmids may lead to the co-selection of *poxtA*, contributing to its persistence and accelerating its dissemination even in the absence of direct selective pressure by the use of phenicols, tetracyclines and oxazolidinones. Attention should be paid to the potential risks of the transfer of the plasmid-borne *poxtA* from enterococci and staphylococci to other Gram-positive bacteria. Therefore, routine surveillance for the spread of *poxtA* in different Gram-positive bacteria and the prudent use of antimicrobial agents in food-producing animals are urgently warranted.

## Figures and Tables

**Figure 1 pathogens-10-00601-f001:**
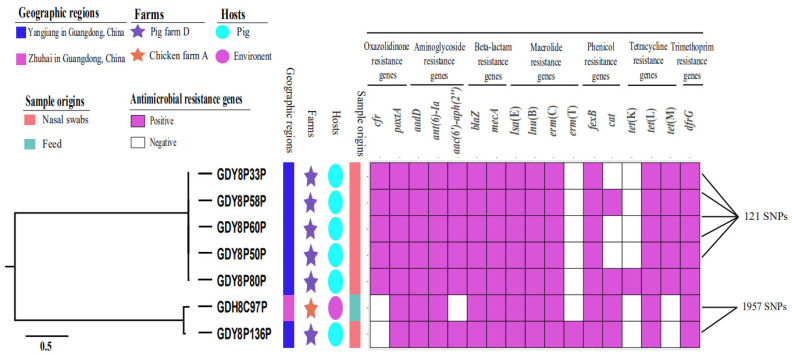
Genomic analysis of five *Staphylococcus haemolyticus* isolates carrying *poxtA* and *cfr* and two *Staphylococcus saprophyticus* isolates carrying *poxtA* of various origins in Guangdong, China. Phylogenic tree was constructed using CSI Phylogeny 1.4. Sources of the isolates are indicated by different colors for geographic regions (squares), farms (stars), hosts (circles) and sample origins (squares). Antimicrobial resistance genes are indicated by the following method: purple, positive; white, negative.

**Figure 2 pathogens-10-00601-f002:**
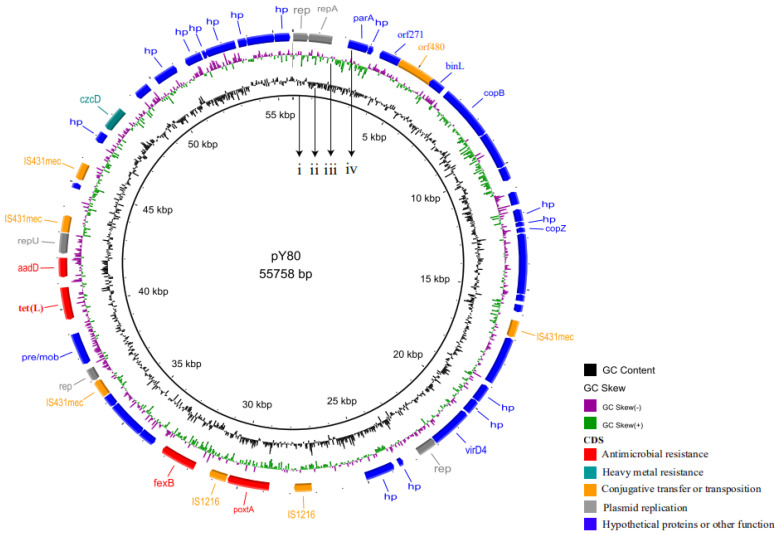
Annotation of plasmid pY80. Circles were displayed (inside to outside) (i) size in bp; (ii) GC content; (iii) GC skew; (iv) positions and directions of predicted coding sequences are indicated by colored arrows according to their predicted functions. Red arrows represent resistance genes, teal arrows represent heavy meatal resistance genes, orange arrows represent genes involved in transfer or transposition, gray arrows represent plasmid replication genes, blue arrows represent genes of unknown functions or other functions.

**Figure 3 pathogens-10-00601-f003:**
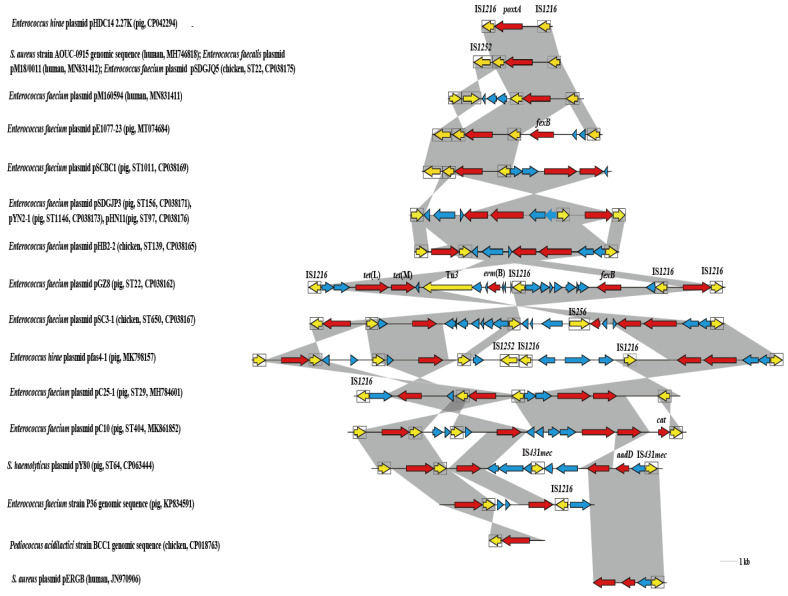
Comparison of the genetic contexts of *poxtA* in plasmid pY80 investigated in this study with corresponding sequences in other plasmids and genomic DNA. Arrows indicate the positions and orientations of the genes. Antimicrobial resistance genes are shown in red. Mobile element regions are underlined in yellow. Insertion sequences are indicated as boxes, with the arrow inside the box showing the transposase gene. Genes with unknown functions and other functions are shown in light blue. Regions of >98% nucleotide sequence identity are shaded grey. Δ indicates an incomplete gene.

**Table 1 pathogens-10-00601-t001:** Background information on the 7 CoNS isolates carrying the *poxtA* gene.

Isolate	Sampling Time	Origin (Farm Type)	Species	MLST
**GDH8C97P**	June 2018	Feed sample (chicken farm A)	*S. saprophyticus*	–
**GDY8P33P**	December 2018	swine nasal swab (pig farm D)	*S. haemolyticus*	ST64
**GDY8P50P**	December 2018	swine nasal swab (pig farm D)	*S. haemolyticus*	ST64
**GDY8058P**	December 2018	swine nasal swab (pig farm D)	*S. haemolyticus*	ST64
**GDY8P60P**	December 2018	swine nasal swab (pig farm D)	*S. haemolyticus*	ST64
**GDY8P80P**	December 2018	swine nasal swab (pig farm D)	*S. haemolyticus*	ST64
**GDY8P136P**	December 2018	swine nasal swab (pig farm D)	*S. saprophyticus*	–

MLST: “–” indicates that *S. saprophyticus* cannot be typed by MLST.

**Table 2 pathogens-10-00601-t002:** Characterization of *poxtA*-positive strains, their electrotransformants and the recipient strain.

Bacterial Isolate	MICs (mg/L)																			Resistance Genes
	AMO	PEN	FOX	GEN	AMI	NEO	KAN	DOX	TET	TIG	FFC	ERY	RIF	VAN	CIP	ENR	LZD	TZD	SXT	
***S. aureus*** **RN4220**	0.125	0.125	2	0.25	1	0.25	0.25	0.125	0.5	0.06	2	0.25	0.008	1	0.5	0.25	0.5	0.06	0.25	-
**GDH8C97P**	8	16	8	0.125	0.25	1	0.25	>64	>64	0.25	>64	>256	0.015	1	4	16	0.5	0.06	0.5	*aadD*, *fexB*, *poxtA*, *tet*(L)
**RN4220/pH97**	0.06	0.125	2	0.125	0.25	0.25	0.25	1	2	0.06	32	>256	0.008	1	0.25	0.125	0.5	0.06	0.25	*fexB*, *poxtA*, *tet*(L)
**GDY8P33P**	2	4	16	8	2	4	8	>64	>64	0.25	>64	>256	0.004	1	>64	32	2	0.06	2	*aadD*, *fexB*, *poxtA*, *tet*(L), *tet*(M)
**RN4220/pY33**	0.06	0.125	2	0.125	2	1	0.25	1	2	0.125	32	0.125	0.008	1	0.5	0.125	0.5	0.06	0.25	*aadD*, *fexB*, *poxtA*, *tet*(L)
**GDY8P50P**	4	4	32	8	2	4	8	64	>64	0.25	>64	>256	0.004	1	>64	>64	2	0.25	8	*aadD*, *fexB*, *poxtA*, *tet*(L), *tet*(M)
**RN4220/pY50**	0.06	0.125	2	0.125	2	1	0.25	2	2	0.125	32	0.125	0.008	1	0.5	0.125	0.5	0.06	0.25	*aadD*, *fexB*, *poxtA*, *tet*(L)
**GDY8058P**	4	4	32	8	2	4	8	64	>64	0.25	>64	>256	0.004	1	>64	>64	4	0.5	16	*aadD*, *fexB*, *poxtA*, *tet*(L), *tet*(M)
**RN4220/pY58**	0.06	0.125	2	0.125	2	1	0.25	2	1	0.125	32	0.125	0.008	1	0.5	0.125	0.5	0.06	0.25	*aadD*, *fexB*, *poxtA*, *tet*(L)
**GDY8P60P**	2	1	32	8	2	4	8	64	64	0.25	>64	>256	0.008	1	>64	>64	2	0.5	8	*aadD*, *fexB*, *poxtA*, *tet*(L), *tet*(M)
**RN4220/pY60**	0.06	0.125	2	0.125	2	1	0.25	1	1	0.25	32	>256	0.002	1	0.5	0.125	0.5	0.06	0.25	*aadD*, *fexB*, *poxtA*, *tet*(L)
**GDY8P80P**	2	4	16	8	2	4	8	64	>64	0.25	>64	>256	0.004	1	>64	>64	4	0.25	16	*aadD*, *fexB*, *poxtA*, *tet*(L), *tet*(M)
**RN4220/pY80**	0.06	0.125	2	0.125	2	1	0.25	1	2	0.25	32	0.125	0.008	1	0.5	0.125	0.5	0.06	0.25	*aadD*, *fexB*, *poxtA*, *tet*(L)
**GDY8P136P**	16	16	16	8	0.5	4	8	32	64	0.25	64	>256	0.03	2	4	2	2	0.25	1	*aadD*, *fexB*, *poxtA*, *tet*(L)
**RN4220/pY136**	0.06	0.125	1	0.125	0.5	0.25	0.25	0.06	2	0.03	16	0.125	0.002	1	0.25	0.125	0.5	0.06	0.25	*fexB*, *poxtA*, *tet*(L)

AMO, amoxicillin; PEN, penicillin; FOX, cefoxitin; GEN, gentamicin; AMI, NEO, neomycin; KAN, kanamycin; amikacin; DOX, doxycycline; TET, tetracycline; TIG, tigecycline; FFC, florfenicol; ERY, erythromycin; RIF, rifampicin; VAN, vancomycin; CIP, ciprofloxacin; ENR, enrofloxacin; LZD, linezolid; TZD, tedizolid; SXT, trimethoprim-sulfamethoxazole. Resistance genes: “–” indicates that no resistance gene was shown in the area. MICs (mg/L) shaded grey represent strains that were resistant to the corresponding antimicrobial agents; despite the lack of clinical breakpoints applicable to staphylococci, the MICs of neomycin and kanamycin were detected in *poxtA*-positive strains, their electrotransformants and the recipient strain *S. aureus* RN4220.

## Data Availability

Not applicable.

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
