# Peer review of "Identification of the Multiresistance Gene poxtA in Oxazolidinone-Susceptible Staphylococcus haemolyticus and Staphylococcus saprophyticus of Pig and Feed Origins"

_pathogens, 2021, doi:10.3390/pathogens10050601_

Round 1
Reviewer 1 Report
Chen and co-worker reported a surveillance regarding the presence of poxtA genes in animal staphylococci in China. They found 7 poxtA-positive isolates, some of them strictly related, and assessed the localization, transferability of the poxtA genes. Moreover, they characterized by WGS a new poxtA-carrying plasmid.
The work is interesting and add new information on spread and genetic background of the poxtA gene. On the other hand, there are some issues in the text that should be carefully addressed prior to recommend the publications of this manuscript.
Major points
- Relatedness of the isolates: why the authors used a phylogenic tree based on PFGE pulsotypes instead of WGS data? PFGE analysis is subjected to several variables that resulted in an altered tree. WGS reads should be used to build the phylogenetic tree (there are a number of tools online, e.g. CSI phylogeny on Center for Genomic Epidemiology)
- Genetic environment of poxtA gene: the real poxtA carrying element is composed by IS1216-poxtA-IS1216. As also highlighted by other papers (3390/microorganisms8122021), this element could be capable of circularization. The analysis of ΔIS1216-poxtA element is worthless (this element resulted probably from an incomplete assembly due to the double IS1216 copies flanking poxtA, that could not be resolved by the assembler considering the short reads obtained with the sequencing). In my opinion this paragraph (as well as the figure 2) should be changed into the analysis of the complete plasmid obtained by Pacbio sequencing, specifying that the plasmid was resolved only in 1 strain.
- Worrisome terms about “poxtA-emergence” through the manuscript. Along the paper the authors described their findings as a big concern (e.g. lines 29-31, 63, 167-168, 193-195, 270-272). I think that their findings are interesting but they should mitigate the sentences regarding the emergence of poxtA isolates (they found 1% of the total strains to be poxtA-positive).
Minor comments
Through the manuscript – change gram-positive in Gram-positive and sensitive in susceptible.
Lanes 43-44 – Remove the abbreviated names into brackets.
Lane 47 – Remove highly.
Lane 49 – Citations should be substituted with a review.
Lanes 66-69: The authors anticipated the results. Remove.
Numbering of results section: check the number and titles of this section (e.g. paragraph 2.2 described also the transferability of poxtA and paragraphs 3.3 and 3.4 had wrong numbers and same titles)
Lane 76: Remove “by PCR analysis”.
Lane 87-89: two transformants are erythromycin resistant.
Lanes 95-97: The sentence is not clear.
Lanes 172-174: check species names.
Lanes 188-190: Mitigate the sentences, the authors find poxtA in the same plasmid.
Table 1 – S. saprophyticus can’t be typed by MLST.
Supplemental – Check captions.
Author Response
Response to Reviewer 1 Comments
Point 1: "Relatedness of the isolates: why the authors used a phylogenic tree based on PFGE pulsotypes instead of WGS data? PFGE analysis is subjected to several variables that resulted in an altered tree. WGS reads should be used to build the phylogenetic tree (there are a number of tools online, e.g. CSI phylogeny on Center for Genomic Epidemiology)".
Response 1: Line 83-84: I have constructed a phylogenic tree using CSI Phylogeny 1.4 instead of PFGE in my manuscript.
Point 2: "Genetic environment of poxtA gene: the real poxtA carrying element is composed by IS1216-poxtA-IS1216. As also highlighted by other papers (3390/microorganisms8122021), this element could be capable of circularization. The analysis of ΔIS1216-poxtA element is worthless (this element resulted probably from an incomplete assembly due to the double IS1216 copies flanking poxtA, that could not be resolved by the assembler considering the short reads obtained with the sequencing). In my opinion this paragraph (as well as the figure 2) should be changed into the analysis of the complete plasmid obtained by Pacbio sequencing, specifying that the plasmid was resolved only in 1 strain".
Response 2: Line 30, 32, 125-126, 133, 135, 142, 145, 150-151, 173, 175-176, 271 and 273: I have replaced IS1216E with IS1216 in the manuscript. In addition, I have removed the analysis of ΔIS1216-poxtA and I finally focused on the analysis of the complete plasmid pY80 from the isolate GDY8P80P in Figure 3.
Point 3: "Worrisome terms about “poxtA-emergence” through the manuscript. Along the paper the authors described their findings as a big concern (e.g. lines 29-31, 63, 167-168, 193-195, 270-272). I think that their findings are interesting but they should mitigate the sentences regarding the emergence of poxtA isolates (they found 1% of the total strains to be poxtA-positive)."
Response 3: I have mitigated the sentences regarding the emergence of poxtA isolates. Line 30: I have replaced "emergence of poxtA" with "poxtA gene". The sentence "The screening of poxtA gene in CoNS isolates from livestock will make big difference" in lines 60-61 was deleted. I have replaced "emergence" with "spread" in lines 279-280.
Point 4: "Through the manuscript – change gram-positive in Gram-positive and sensitive in susceptible."
Response 4: I have replaced "gram-positive" with "Gram-positive" in lines 31, 45-46, 177 and 279-280. I have replaced "sensitive "with "susceptible" in lines 101-102.
Point 5: "Lanes 43-44 – Remove the abbreviated names into brackets."
Response 5: I have removed the abbreviated names into brackets in the lines 42-43.
Point 6: "Lane 47 – Remove "highly"
Response 6: I have removed "highly" in the line 45.
Point 7: "Lane 49 – Citations should be substituted with a review."
Response 7: I have replaced the citations with a review in the line 47.
Point 8: "Lanes 66-69: The authors anticipated the results. Remove."
Response 8: I have removed the content in the lines 63-64.
Point 9: "Numbering of results section: check the number and titles of this section (e.g. paragraph 2.2 described also the transferability of poxtA and paragraphs 3.3 and 3.4 had wrong numbers and same titles)"
Response 9: I have checked the number and titles, and the wrong number and titles were corrected in paragraph 3.3, 3.4, 3.5 and 3.6 in lines 97-132.
Point 10: "Lane 76: Remove "by PCR analysis".
Response 10: I have removed "by PCR analysis" in the line 69.
Point 11: "Lane 87-89: two transformants are erythromycin resistant."
Response 11: I have replaced "The seven poxtA-positive electrotransformants were sensitive to all tested antibiotics except erythromycin and florfenicol" with "Two electrotransformants are erythromycin resistant" in the line 104.
Point 12: "Lanes 95-97: "The sentence is not clear."
Response 12: I have replaced " In addition, the coexistence of cfr, aac(6')-aph(2'') and tet(M) genes in the poxtA-positive S. haemolyticus isolates was confirmed " with " For example, the cfr, tet(M) and aac(6')-aph(2'') genes were also identified in the poxtA-positive S. haemolyticus isolates " in lines 78-79.
Point 13: "Lanes 172-174: check species names."
Response 13: I have corrected the wrong species names in the lines 166-172.
Point 14: "Lanes 188-190: Mitigate the sentences, the authors find poxtA in the same plasmid."
Response 14: I have replaced "Plasmid pY80 showed low coverage (<38%) with other plasmids in the NCBI database, indicating that poxtA was at the risk of spreading to other new plasmids" with "The poxtA gene was identified in the new plasmid" in line 186.
Point 15: "Table 1 – S. saprophyticus can’t be typed by MLST."
Response 15: I have corrected the wrong content in Table 1 in lines 70-71.
Point 16: "Supplemental – Check captions."
Response 16: I have replaced "GDH8C90P" with "GDH8C97P" in the Supplementary.

Reviewer 2 Report
Line 48 and elsewhere: Gram-positive with a capital G (as it refers to a name)
Lin 51: As far as OI know, the role of mutations in the gene for the ribosomal protein L22 in oxazolidinone resistance has not been confirmed.
Line 67: … carry the cfr gene …
Table 1: It would be more helpful for the reader to split table 1 in two tables: one with the backround data, such as sampling time, origin, species and MLST and the other with the MICs. In this second table, it would be helpful to include the resistance genes present with those located on the poxt A-carrying transferred plasmid in bold or underlined. This information is missing when comparing the MIC profiles of the original strain and its transformant
Table 1 is difficult to read – maybe the font can be decreased so that the abbreviation of antimicrobial agents are in the same line.
Table 1: The authors should test all antimicrobial agents to which resistance genes are located on poxtA-carrying plasmids, i.e. kanamycin and neomycin for aadD. I would also test tetracycline in addition to doxycycline. For czcD, at least zinc should also be tested.
Table 1 GDH8C97P: How can It be that the MIC for amikacin is 0.125 in the original strain and 2 in the transformant
Table 1/Figure 2: Why has the S. aureus transformant carrying plasmid pY80 with a tet(L) gene only a doxycycline MIC of 1 which also classifies this transformant as tetracycline-susceptible?
Tabble 1: When applying CLSI clinical breakpoints, all staphylococci listed in Table 1 are susceptible to linezolid (S<= 4 mg/L, R >= 8 mg/L) and tedizolid (S <= 0.5 µg/L) … even those isolates that carry two oxazolidinone resistance genes cfr + poxtA.
Line 82: Antimicrobial susceptibility testing
Line 82: resistance rates
Line 83: cipro-floxacin
Line 85, lines 88-89 and elsewhere: susceptible instead of sensitive
Lines 87-88: S, aureus in italics
Line 89: antimicrobial agents instead of antibiotics
Figure 1: Part of this figure is only readable if enlarged 400- to 600-fold. This is not acceptable, i.e. far too small for the printed version. Moreover several genes are not correctly displayed: lsa(E), erm(C), erm(T) with lsa and erm in italics and E, C, T in brackets and not in italics.
Line 114: … located on plasmids …
Figure 2: Is the Tn552 in Figure 1 the complete transposon? If so, it would ne netter to include the different genes within Tn552. If not, state clearly which Tn552-related genes are present and if they are complete.
Database entry of plasmid pY80: In the database entry, the genes fexB and czcD are not annotated and included. This needs to be corrected.
Figure 3: This figure is also far too small for the printed version. To be able to read it and see the details, it needs to be enlarged at least 400-fold.
Discussion: The authors need to address and discuss the observation that the strains and transformants described in their study are phenotypically oxazolidinone-susceptible despite the fact that they carry up to two oxazolidinone resistance genes. The lack of phenotypic tetracycline resistance in transformants carrying plasmid pY80 also needs to be discussed.
Line 224: gentamicin
Line 253: sequence of plasmid …
Author Response
Point 1: "Line 48 and elsewhere: Gram-positive with a capital G (as it refers to a name)"

Response 1: I have replaced "gram-positive" with "Gram-positive" in lines 31, 45-46, 177 and 279-280.
Point 2: "Lin 51: As far as OI know, the role of mutations in the gene for the ribosomal protein L22 in oxazolidinone resistance has not been confirmed."
Response 2: I have replaced "L3, L4 and L22" with "L3 and L4".
Point 3: "Line 67:… carry the cfr gene …"
Response 3: I have removed the content in the lines 63-64 according to the reviewer 1.
Point 4: "Table 1: It would be more helpful for the reader to split table 1 in two tables: one with the background data, such as sampling time, origin, species and MLST and the other with the MICs. In this second table, it would be helpful to include the resistance genes present with those located on the poxtA-carrying transferred plasmid in bold or underlined. This information is missing when comparing the MIC profiles of the original strain and its transformant".
Response 4: I have spilt table 1 in two tables: one with the background data, such as sampling time, origin, species and MLST and the other with the MICs and the resistance genes present with those located on the poxtA-carrying transferred plasmids in lines 70-71 and 91-96. I have added the comparison of the MIC profiles of the original strain and its transformant in lines 104-107.
Point 5: "Table 1 is difficult to read – maybe the font can be decreased so that the abbreviation of antimicrobial agents is in the same line."
Response 5: The font in Table 2 in lines 91-96 was decreased to make the abbreviation of antimicrobial agents in the same line.
Point 6: "Table 1: The authors should test all antimicrobial agents to which resistance genes are located on poxtA-carrying plasmids, i.e. kanamycin and neomycin for aadD. I would also test tetracycline in addition to doxycycline. For czcD, at least zinc should also be tested."
Response 6: I have added the antimicrobial susceptibility testing of the original strains and transformants to kanamycin, neomycin and tetracycline in lines 229-230. I feel very sorry that I cannot complete the test of zinc, because our laboratory has been engaged in the study of bacterial resistance and we have no experience in testing heavy metal.
Point 7: "GDH8C97P: How can It be that the MIC for amikacin is 0.125 in the original strain and 2 in the transformant".
Response 7: I have repeated the antimicrobial susceptibility testing of the strain GDH8C97 and corresponding transformant to amikacin, and I have corrected the wrong contents in Table 2 in lines 91-96.
Point 8: "Table 1/Figure 2: Why has the S. aureus transformant carrying plasmid pY80 with a tet(L) gene only a doxycycline MIC of 1 which also classifies this transformant as tetracycline-susceptible?"
Response 8: In our study, the recipient strain S. aureus RN4220 carrying plasmids with tet(L) gene did not show resistance to doxycycline and tetracycline in lines 91-96. It might be related to the silencing of the tet(L) gene.
Point 9: "Table 1: When applying CLSI clinical breakpoints, all staphylococci listed in Table 1 are susceptible to linezolid (S<= 4 mg/L, R >= 8 mg/L) and tedizolid (S <= 0.5 µg/L) … even those isolates that carry two oxazolidinone resistance genes cfr + poxtA."
Response 9: The results showed that those isolates in Table 2 were susceptible to linezolid and tedizolid in lines 91-96.
Point 10: "Line 82: Antimicrobial susceptibility testing"
Response 10: I have replaced "Antibiotic susceptibility testing" with "Antimicrobial susceptibility testing" in the line 98.
Point 11: "Line 82: resistance rates"
Response 11: I have replaced "resistant rates " with "resistance rates" in the line 98.
Point 12: "Line 83: cipro-floxacin"
Response 12: The wrong name was corrected in the line 100.
Point 13: "Line 85, lines 88-89 and elsewhere: susceptible instead of sensitive".
Response 13: I have replaced "sensitive "with "susceptible" in lines 101-102.
Point 14: "Lines 87-88: S, aureus in italics".
Response 14: I have made S. aureus italic in the line 104.
Point 15: "Line 89: antimicrobial agents instead of antibiotics".
Response 15: I have replaced "The seven poxtA-positive electrotransformants were sensitive to all tested antibiotics except erythromycin and florfenicol" with "Two electrotransformants are erythromycin resistant" according to the reviewer 1.
Point 16: "Figure 1: Part of this figure is only readable if enlarged 400- to 600-fold. This is not acceptable, i.e. far too small for the printed version. Moreover several genes are not correctly displayed: lsa(E), erm(C), erm(T) with lsa and erm in italics and E, C, T in brackets and not in italics."
Response 16: I have improved the resolution of figure 1 and I have corrected the names of lsa(E), erm(C) and erm(T) of figure 1 in lines 83-84.
Point 17: "Line 114: … located on plasmids …".
Response 17: I have replaced " located in plasmids " with " located on plasmids " in the line 117.
Point 18: "Figure 2: Is the Tn552 in Figure 1 the complete transposon? If so, it would ne netter to include the different genes within Tn552. If not, state clearly which Tn552-related genes are present and if they are complete."
Response 18: The Tn552 is the complete transposon in Figure 2. According to the plasmid pAFS11 (GenBank: FN806789) in the NCBI database, the CDS should be annotated with Tn552. Actually, the transposon Tn552 contained 8 bp of ATP-binding protein and 29 bp of DNA-invertase.
Point 19: "Database entry of plasmid pY80: In the database entry, the genes fexB and czcD are not annotated and included. This needs to be corrected."
Response 19: I have e-mailed to the NCBI staff for the correction.
Point 20: "Figure 3: This figure is also far too small for the printed version. To be able to read it and see the details, it needs to be enlarged at least 400-fold."
Response 20: I have improved the resolution of figure 3.
Point 21: "Discussion: The authors need to address and discuss the observation that the strains and transformants described in their study are phenotypically oxazolidinone-susceptible despite the fact that they carry up to two oxazolidinone resistance genes. The lack of phenotypic tetracycline resistance in transformants carrying plasmid pY80 also needs to be discussed."
Response 21: I have added the discussion about the observation that the strains and transformants described in their study are phenotypically oxazolidinone-susceptible despite the fact that they carry up to two oxazolidinone resistance genes and the lack of phenotypic tetracycline resistance in transformants carrying plasmid pY80 in lines 193-201 in our manuscript.
Point 22: "Line 224: gentamicin".
Response 22: I have replaced "gentamycin" with "gentamicin" in the line 229.
Point 23: "Line 253: sequence of plasmid …".
Response 23: I have replaced "sequences of plasmid" with "sequence of plasmid" in the line 262.
Round 2
Reviewer 1 Report
The authors answered all my questions and the manuscript is now considerely improved.
I have just two point.
The citated review on line 50 is not appropriated. I suggest to cite papers on oxazolidinone resistance mechanisms. For example you can cite the recent review from Bender et al. (10.1016/j.drup.2018.10.002).
In the new table 2 the authors should also insert the resistance mechanisms from donors strains, to better understand what type of gene are trasmitted by transformation.
Author Response
Point 1: "The citated review on line 50 is not appropriated. I suggest to cite papers on oxazolidinone resistance mechanisms. For example, you can cite the recent review from Bender et al. (10.1016/j.drup.2018.10.002)."
Response 1: I have cited the recent review according to your advice on line 50.
Point 2: " In the new table 2 the authors should also insert the resistance mechanisms from donors strains, to better understand what type of gene are trasmitted by transformation."
Response 2: I have included the tested resistance genes present in the original strains in the new table 2.
Reviewer 2 Report
-
Title: I strongly suggest to change the title into ‘Identification of the multiresistance gene poxtA in oxazolidinone-susceptible Staphylococcus haemolyticus and Staphylococcus saprophyticus of pig and feed origins’
-
Line 79: The opening bracket in tet(M) not in italics
-
Line 91 ff: Table 2 - The authors also need to include the resistance genes present in the original strains
-
Line 91 ff: Table 2 - The authors may consider placing the line with the S. aureus RN4220 MICs at the top of Table 2 instead of somewhere in the middle.
-
The database entry has not been corrected.
-
Lines 105/106: Please delete ‘similar or’
-
Line 106: Please delete gentamicin as none of these genes confers resistance to gentamicin
-
Fig. 1: Plasmid pAFS11 is not the correct reference for Tn552. This plasmid also carries only part of Tn552. Please refer to Genbank accession no. X52734.1 for the complete Tn552. Please modify Fig. 1 accordingly.
-
Line 198: Replace gentamicin by kanamycin and neomycin.
-
Line 201 and elsewhere in the text: Please replace antibiotics by antimicrobial agents.
Author Response
Point 1: "Title: I strongly suggest to change the title into ‘Identification of the multiresistance gene poxtA in oxazolidinone-susceptible Staphylococcus haemolyticus and Staphylococcus saprophyticus of pig and feed origins."
Response 1: I have changed the title according to your advice on lines 2-4.
Point 2: "Line 79: The opening bracket in tet(M) not in italics."
Response 2: I have made the opening bracket in tet(M) italic on line 79.
Point 3: " Line 91 ff: Table 2 - The authors also need to include the resistance genes present in the original strains."
Response 3: I have included the tested resistance genes present in the original strains in the new table 2.
Point 4: "The database entry has not been corrected."
Response 4: I have checked my e-mail and it will take a few days for NCBI staff to modify it.
Point 5: " Lines 105/106: Please delete ‘similar or’."
Response 5: I have removed ‘similar or’ on line 105.
Point 6: "Line 106: Please delete gentamicin as none of these genes confers resistance to gentamicin."
Response 6: I have removed " gentamicin " on line 106.
Point 7: "Fig. 1: Plasmid pAFS11 is not the correct reference for Tn552. This plasmid also carries only part of Tn552. Please refer to Genbank accession no. X52734.1 for the complete Tn552. Please modify Fig. 1 accordingly. "
Response 7: I referred to Genbank accession no. X52734.1 for the complete Tn552 according to your advice. I found the Tn552 is not the complete transposon in Figure 2 and the genes of orf271, orf480 and binL are part of Tn552. The orf271 gene encodes ATP-binding protein, the orf480 gene encodes transposase.
Point 8: " Line 198: Replace gentamicin by kanamycin and neomycin."
Response 8: I have replaced gentamicin by kanamycin and neomycin on lines 198-199.
Point 9: " Line 201 and elsewhere in the text: Please replace antibiotics by antimicrobial agents."
Response 9: I have replaced antibiotics by antimicrobial agents on lines 201 and 280-281.